# Cellular Plasticity: A Route to Senescence Exit and Tumorigenesis

**DOI:** 10.3390/cancers13184561

**Published:** 2021-09-11

**Authors:** Hadrien De Blander, Anne-Pierre Morel, Aruni P. Senaratne, Maria Ouzounova, Alain Puisieux

**Affiliations:** 1Equipe Labellisée Ligue Contre le Cancer “EMT and Cancer Cell Plasticity”, CNRS 5286, INSERM 1052, Centre Léon Bérard, Cancer Research Center of Lyon, Université Claude Bernard Lyon 1, 69008 Lyon, France; anne-pierre.morel@lyon.unicancer.fr (A.-P.M.); maria.ouzounova@lyon.unicancer.fr (M.O.); 2LabEx DEVweCAN, Université de Lyon, 69008 Lyon, France; 3Institut Curie “EMT and Cancer Cell Plasticity”, Consortium Centre Léon Bérard, 69008 Lyon, France; 4UMR3664—Nuclear Dynamics, Development, Biology, Cancer, Genetics and Epigenetics, Institut Curie, PSL Research University, 75005 Paris, France; aruni.senaratne@curie.fr; 5CNRS UMR3666, Inserm U1143, Cellular and Chemical Biology, Institut Curie, PSL Research University, 75005 Paris, France

**Keywords:** cellular plasticity, senescence, reprogramming, immune evasion, epithelial-mesenchymal transition

## Abstract

**Simple Summary:**

Senescence is a form of cell cycle arrest induced by stresses such as DNA damage and oncogenes and therefore constitutes a crucial barrier against cancer. Nevertheless, senescent cells can escape or bypass this tumor suppressor mechanism and evolve towards an altered, pre-cancerous genotype. Furthermore, accumulated senescent cells that are not cleared by the immune system secrete pro-inflammatory factors, promoting malignant phenotypes. This pro-tumor activity of senescence is associated with genetic reprogramming and the acquisition of cellular plasticity. In this review, we aim to unravel the interconnection between senescence, senescence-associated pro-inflammatory cytokines and the induction of cellular plasticity, which enables the adaptability of tumor cells at different stages of carcinogenesis.

**Abstract:**

Senescence is a dynamic, multistep program that results in permanent cell cycle arrest and is triggered by developmental or environmental, oncogenic or therapy-induced stress signals. Senescence is considered as a tumor suppressor mechanism that prevents the risk of neoplastic transformation by restricting the proliferation of damaged cells. Cells undergoing senescence sustain important morphological changes, chromatin remodeling and metabolic reprogramming, and secrete pro-inflammatory factors termed senescence-associated secretory phenotype (SASP). SASP activation is required for the clearance of senescent cells by innate immunity. Therefore, escape from senescence and the associated immune editing would be a prerequisite for tumor initiation and progression as well as therapeutic resistance. One of the possible mechanisms for overcoming senescence could be the acquisition of cellular plasticity resulting from the accumulation of genomic alterations and genetic and epigenetic reprogramming. The modified composition of the SASP produced by these reprogrammed cancer cells would create a permissive environment, allowing their immune evasion. Additionally, the SASP produced by cancer cells could enhance the cellular plasticity of neighboring cells, thus hindering their recognition by the immune system. Here, we propose a comprehensive review of the literature, highlighting the role of cellular plasticity in the pro-tumoral activity of senescence in normal cells and in the cancer context.

## 1. Introduction

Senescence is characterized by permanent cell cycle arrest and is generally considered irreversible. It is triggered by developmental signals or various endogenous and exogenous stressors, including telomere shortening, physically or chemically induced DNA damage, oncogene activation, exposure to reactive oxygen species (ROS), and therapeutic treatments like chemo- or radiotherapies [1]. The cessation of cell proliferation via senescence involves a number of phenotypic and functional changes to cells. These include morphological alterations, chromatin remodeling, metabolic reprogramming and secretion of a complex mix of proinflammatory factors termed the senescence-associated secretory phenotype (SASP) [2]. Depending on the type of senescence-inducing trigger and the type of target cell, the collective changes accompanying individual senescence responses are highly diverse. For example, differences have been observed in the stop phase of the cell cycle as well as in the types of regulatory circuits involved during senescence of fibroblasts vs. senescence of epithelial cells, keratinocytes or melanocytes [3,4].

The senescence pathway has critical implications in normal physiological processes like embryonic development, tissue regeneration and wound healing [5,6,7,8]. For example, fibroblast cells, which play major roles in maintaining tissue structure, will senesce at a wound site to release SASP factors that permit their differentiation into myofibroblasts, the main repair cells that restore tissue integrity and facilitate wound closure following injury [5,9,10]. The senescence program is thus generally beneficial, involving a positive, transient role for senescent cells. However, if these cells persist, then senescence plays a causal and deleterious role in certain pathologies. For instance, the buildup of senescent fibroblasts in tissues and organs is central to age-related disruption of tissue structure and function and conditions like fibrosis [11,12,13]. Accumulation of senescent cells is also implicated in cancer [7,14,15,16]. Indeed, two outcomes as disparate as wound healing vs. tumorigenesis, consolidated by one biological pathway, supports the theory that a tumor is a ‘wound that does not heal’ [17,18,19].

Although the vast majority of studies on senescence have been carried out on fibroblast models, another cell type that is highly relevant to understanding the deteriorative effects of the senescence program is the epithelial cell, another key contributor to tissue and organ structure and function. Epithelial cells can undergo cellular reprogramming through epithelial to mesenchymal transition (EMT), a process that imparts features of stemness and plasticity to cells [20,21,22]. While EMT, like senescence, is physiologically involved in development and wound repair [23,24], dysfunctional EMT associated with the senescence program is a major contributor to certain pathologies. For example, epithelial cells morphologically close to myofibroblasts can undergo EMT, leading to epithelial remodeling and massive extracellular matrix (ECM) deposition, ECM stiffness and the thickening of pulmonary tissue resulting in fibrosis [25,26]. These cells, reprogrammed during chronic inflammation, are a key component of the senescence response, and their aberrant accumulation further catalyzes the generation of damaged fibrotic tissue [25,26,27,28,29].

The interplay between cellular reprogramming and senescence has drastic implications for tumor development and progression [30,31,32,33,34,35]. In this context, malignant cells can undergo EMT-mediated reprogramming to acquire features of plasticity and stemness that help these cells overcome the tumor suppressor action of senescence and continue proliferating [36,37,38,39,40,41,42]. Specifically, cells can defeat the senescence barrier either through a mechanism of “escape” or a mechanism of “bypass”. The molecular context of these two mechanisms will be described in detail in the next sections. Senescence is thus observed to have opposite effects in the circumstances of cancer; first, in preventing the proliferation risk of neoplastic cells via cell cycle arrest; yet, second, in creating a window of opportunity for cancer cells to fight senescence and progress along the path to more aggressive tumors. In this review, we elucidate this double-edged sword in cancer biology by deciphering how the interplay between senescence and cellular plasticity has an impact from tumor initiation up to therapeutic resistance.

## 2. Overcoming Senescence via *Escape* Instigates Neoplasticity and Genomic Instability in Pre-Tumoral Cells

Senescence is considered an “evolutionary cul-de-sac” for cells, as it generally implies the inability to resume cell proliferation [36,43,44,45]. However, some cells like epithelial cells are capable of spontaneously reverting to a proliferative state following cell cycle arrest after encountering a stress signal, a process known as “senescence escape”. Senescence escape is facilitated by altering the activity of chromatin regulators, metabolic pathways or extracellular pH [43,46,47]. The senescence program in epithelial cells involves a period known as “stasis”, or telomere-independent senescence. Here, a decrease in poly (ADP) ribose-polymerase 1 (PARP1) expression compromises the repair of single strand breaks (SSBs) in senescent epithelial cells, which induces the persistence of single strand break repair (SSBR) foci [4,48]. In turn, a signaling cascade is engaged, leading to p38MAPK/p16-mediated cell cycle arrest (stasis). These events are associated with some characteristic features: enlarged and flattened morphology, an inflammatory secretome, and senescence-associated β-galactosidase-positive, polynucleated cells [48,49,50,51,52,53].

It has been estimated that about one in 10^4^ epithelial cells spontaneously escapes from stasis and reenters the cell cycle to give rise to new clones [50,54,55]. This is associated with certain epigenetic changes that distinguish “escaped” epithelial cells from the rest, including hypermethylation of p16 promoter DNA, resulting in decreased CDKN2A expression [56,57,58,59,60]. Notably, the reported epigenetic modifications have been correlated with methylation signatures found in hyperplasia, which is often the initial stage of cancer development [57].

Escape from stasis in epithelial cells is found to accompany the acquisition of some hallmark features of cancer cells. Notably, cell plasticity is induced through EMT, as evident in the expression of EMT transcription factors (EMT-TFs) [48]. Furthermore, epithelial and mesenchymal markers are found to be lost and gained, respectively, which is known to promote invasive and metastatic properties of cells [48,61,62]. Although the role of EMT in stasis escape has not yet been elucidated, the latter observation suggests that plasticity features may help epithelial cells to overcome oncosuppressive barriers. Of note, given that PARP1 downregulation promotes stasis, EMT has recently been linked to PARP1 decrease as a means of escaping senescence [63,64,65]. In the context of deciphering this connection, it will be interesting to impede EMT or PARP-1 action in epithelial cells and investigate its impact on the stasis escape process as a whole.

While stasis is essentially a characteristic of epithelial cells, three additional modes of senescence are described in other cell types: (i) replicative senescence (RS) or telomere-dependent senescence, which has been historically characterized in fibroblasts and is caused by telomere shortening; (ii) oncogene-induced senescence (OIS), following the activation of an oncogene such as *RAS*; and (iii) therapy-induced senescence (TIS), following cancer treatments like chemo- and radiotherapy [16,44,66]. These senescence types are activated by the p53/p21^WAF1^ tumor suppressor pathway and share the common feature of being induced by genotoxic stresses. Consequently, RS, OIS and TIS are associated with endoreplication, polyploidy-induced DNA damage and genomic instability, as well as extensive epigenetic reprogramming [55,56,59,67,68,69,70,71,72,73]. However, mechanisms of escape have been described that enable these damaged cells to re-enter the cell cycle, including for senescent fibroblasts, which are known to be mostly incapable of resuming cell proliferation owing to the high stability of RS-induced DNA damage [4]. These mechanisms target polyploid cells for depolyploidization and budding.

Polyploid cells are generally considered to be terminally differentiated because they can no longer divide. However, genomic instability in polyploid cells might provide a route to aneuploidy, which is long thought to play a major role in p53 status-independent tumor initiation [73,74,75,76,77,78,79]. Here, multinucleate giant polyploid cells can restore their proliferative capacity by undergoing an atypical type of cell division known as “neosis”, initially identified during tumor progression as well as in normal cells [49,80,81,82,83]. Neosis results in daughter cells with reduced cell ploidy (depolyploidization) and extended mitotic life span [80], thus being at the origin of senescence escape [77,80,82,84,85,86,87]. Like epithelial cells that escape stasis, stem-like properties following depolyploidization have been reported in fibroblasts that have escaped from RS or in melanocytes that have escaped from OIS, allowing the acquisition of anoikis resistance and tumorigenic capacities [77,88]. Furthermore, in fibroblasts undergoing RS, telomere-driven chromosome instability displays massive changes in expression of microRNAs (miR), including the miR-200-family. miR-200 is a well-known negative regulator of EMT-TFs that confers plasticity-related phenotypic traits to immortalized pre-tumoral cells [68].

Senescence escape can thus be facilitated in major cell types by the dedifferentiation associated with large-scale cellular reprogramming and polyploidy. Reminiscent of a blastomere-like process of dedifferentiation in somatic cells for tumor initiation, senescence-escaped cells too can evolve towards novel tumorigenic states which, in the case of polyploidy, can accompany modified genomic profiles [86,89,90].

## 3. Overcoming Senescence via Bypass Preserves a Stable Tumor Genome

At times, cells can continue to proliferate even in the presence of stressors that would likely lead to cell cycle arrest. This process, known as senescence bypass, differs from senescence escape in that a senescent stage is entirely absent in cells. Notably, the propensity of a cell to bypass senescence appears to depend on its differentiation state. More specifically, in contrast to progenitor cells that are more committed to differentiation, stem cells of the young adult do not senesce [91,92,93]. This correlation between the degree of cellular differentiation and the likelihood of bypassing senescence has significant implications in tumor development [41].

As described in the concept of cellular pliancy, each stage of differentiation within a specific cell lineage is associated with a unique susceptibility to malignant transformation when subjected to a specific oncogenic insult [41]. In this context, differentiated cells are typically vulnerable to OIS, whereas stem cells can overcome OIS and DNA damage. This resistance is conferred to stem cells through the expression of EMT-TFs that promote cell plasticity while inhibiting the p16/Rb pathway to cell cycle arrest [34,93]. Specifically, TWIST proteins have been shown to down-regulate the expression of *p16^INK4A^*, *p19^ARF^* and *p21^WAF1^*, thus attenuating p53 responses and allowing cancer cells to escape RAS-induced senescence in fibroblasts and human mammary epithelial cells [34]. ZEB1 was also shown to suppress *p16^INK4A^* and *p15^INK4B^* expression and enable the overcoming of OIS that is triggered by the overexpression of EGFR (epidermal growth factor receptor) in human esophageal epithelial cells [94]. Moreover, the high expression of EMT-TF ZEB1 in mammary stem cells partly prevents oncogene-induced cell stress, in turn allowing the evasion of oncosuppressive barriers and subsequent malignant transformation. Thus, a stem state can allow a cell to continue to proliferate via senescence bypass, while fending off high DNA damage and genome instability that would typically be acquired during a senescent period [95].

## 4. A Dualistic Model for Tumor Initiation

Together, the modes of action of senescence escape and senescence bypass consolidate a dualistic model for overcoming cell cycle arrest and initiating a tumor (Figure 1). More specifically, the escape from senescence of differentiated cells like fibroblasts requires the acquisition of polyploidy and genomic instability. In contrast, stem cells and undifferentiated cells have the intrinsic propensity to bypass senescence, which relies, at least partly, on the cells’ EMT-dependent features of plasticity and is accompanied by the preservation of a normal, stable genome [41,95]. As a result, genomically-stable tumors originate from cells that are able to bypass senescence and evade the onset of genomic instability as observed in tumors originating from senescence-escaped cells [41,95,96]. This difference in the mode of tumor initiation further highlights the relevance of the cell of origin in cancer for conditioning entry into senescence and the necessity for escape [96,97]. In view of this, cells of stem or embryonic origin, which largely include pediatric tumors, are able to bypass senescence [41,95,98].

## 5. SASP: A Major Determinant of Tumorigenesis

The SASP refers to the secretome of senescent cells [2]. As a whole, the SASP comprises a range of substances, including inflammatory cytokines like IL-6 and IL-8, immune modulators, paracrine factors, growth factors and enzymes [14,99]. The SASP is typically induced following chemo- or radiotherapy [100]. However, various other types of stress stimuli, including oncogene activation and telomere attrition, can induce the SASP. Depending on the type of stress and type of cell, the SASP varies in composition and over time to produce a unique, cell-specific response [3,5,101,102,103,104,105].

It has been recently proposed to group SASPs into two main types, despite some redundancy in composition: (i) the major inflammation regulator, NF-KB-dependent inflammatory-type SASP (NASP), under the control of the key inflammation mediator, the cGAS/STING/p38/NFKB/IL-1α axis [102,105,106,107,108,109]; and (ii) the major tumor suppressor, p53-dependent SASP (PASP) (Table 1) [100,107,110]. Even though both types can be found in the same cell population, a dichotomy has indeed been proposed between the two [100,103,110]. Importantly, while the PASP is linked to mitochondrial dysfunction [103], the NASP is found to have tumorigenic properties, thus suggesting a causal link between chronic inflammation and tumor development [110]. Notably, this link is dependent on the paracrine action of NASP factors, which can impact the overall fitness of an entire cell population as compared to the fate of a single cell. For example, OIS-dependent cellular reprogramming via the NASP can lead to population level cellular immortality and carcinogenesis [33,41,111,112,113], as is found to be important for tumor initiation in transgenic models [114], although seemingly not required for tumor initiation in xenografts [107]. Furthermore, the SASP produced by senescent fibroblasts promotes the cancerous development of nearby epithelial cells through paracrine action and is associated with an increase in epithelial cell proliferation and EMT-mediated plasticity [14,115,116,117,118]. While these examples demonstrate the pro-tumor role of the SASP via a paracrine effect, there are few studies that functionally demonstrate a potential role for the SASP in tumorigenesis associated with the escape from or bypass of senescence.

Importantly, the NASP is also strongly linked to innate immunity through the cGAS/STING/NFKB pathway, notably during wound healing [5,117,119]. Here, the recruitment of specialized immune cells via the SASP to a wound site leads to the clearance of senescent cells and thereby shuts off opportunities for these cells to overcome senescence and initiate tumorigenesis (detailed in the next section) [105,107,108,109].

Thus, in addition to its population-scale pro-tumor effects, the SASP is also involved in anti-tumorigenesis at early stages by targeting individual senescent cells that could potentially become tumorigenic [107,118,120]. Many studies report divergent conclusions as to the anti- or pro-tumor role of the SASP [121]. These discrepancies are often attributed to a difference in cell type, stimulus, environmental context or tumor stage [122]. It is also possible that these conclusions depend on the cellular or population-based context, where consideration of the number of cells involved in the SASP is critical for interpretations [123,124].

## 6. Acquisition of Plasticity Allows Immunoevasion by Senescent Cells

Through the SASP, senescent cells recruit immune cells into tissues and bring about their own elimination. Particularly in the non-pathological context of wound healing, senescent cells present at a wound site secrete chemokines like CCL2 as part of the inflammatory secretome, which can attract immature monocytes expressing the chemokine receptor CCR2. These monocytes then become activated into polarized M1 (inflammatory) macrophages that produce interleukins such as IL-1alpha, IL-1beta and IL-6 and allow the recruitment of NK cells to the wound site [120,124,125,126]. NK cells will subsequently recognize and eliminate senescent cells [126,127,128,129]. This process of SASP-mediated immune clearance of senescent cells is also enabled by antigen-presenting cells, which play an important role in the activation of T4 lymphocytes under the influence of M1-secreted cytokines [118]. Senescent cells can further promote their elimination by becoming immunogenic through the expression of stimulatory ligands such as MHC Class I chain-related proteins A and B (MICA/B). MICA/B can bind to the transmembrane receptor NKG2D and activate the killing of senescent cells through the action of NK cells [127,129].

This active role of the immune system in clearing senescent cells means that any impairment of the immune system would result in the persistent accumulation of these cells. Indeed, impaired innate and adaptive immune responses have been reported to result in the increment of various senescent cells, which would become deleterious in the context of tumorigenesis [130,131]. Although not completely validated, the acquisition of plasticity by senescent cells has been described to allow immune evasion, for example, after overcoming polyploidy in fibroblasts [77]. This has made it possible to put forward an association between stemness and immune evasion [132,133]. However, the stemness state does not intrinsically facilitate immune evasion by senescent cells. Rather, it is due to a downregulation of the antigen presentation pathway as a result of a decrease in replication in quiescent stem cells, which in turn prevents their immunoediting [134,135]. Indeed, studies confirm that, in stem cells, which are a key raw material for tumor initiation (described in previous section), quiescence promotes evasion from the immune system in both physiological [134] and pathological [136,137,138] contexts. For example, the lack of MHC class I-mediated antigen presentation on the surface of cancer cells is linked to the level of DNA replication and is shown to generate a stress response that enables these MHCI^−^ cancer cells to evade the immune system and establish metastatic foci [137,139]. Other modes of immune evasion by senescent cells include alterations to the secretory phenotype during senescence escape. These changes in the composition of the secretome (e.g., overexpression of MMP3, which cleaves activating MICA ligands from the senescent cell surface, or HLA-E-, an inhibitory ligand that blocks NK cell killing) can lead to a redirection of macrophage polarization from a tumor-inhibiting M1 state to a tumor-promoting M2 state that prevents the elimination of senescence-escaped cancer cells [124,130,131]. Furthermore, EMT can also lead to reduced immunoediting through the modification of the nature and quantity of antigens presented on the surface of senescent cells due to a downregulation in the expression of immune receptors like MHC-I [140,141] (Figure 2).

## 7. Senescence Escape Is a Driver of Tumor Resilience

Evasion or escape from therapy-induced senescence (TIS) has been reported in several cancer cases [144,145]. In fact, cancer cells that spontaneously escape senescence have been found to have genomic alterations, notably, a polyploid phenotype characterized by the presence of polyploid giant cells [56,146,147,148]. These observations are in line with reports of approximately one-third of tumors being polyploid [149,150], while additional studies carried out both in vitro and in vivo further demonstrate polyploidy in cancer cells [85,151,152,153,154,155]. Thus, similar to pre-tumoral senescent cells (refer Section 1), polyploidy has been described as a means of senescence escape in cancer cells [152].

Importantly, by facilitating escape, the polyploid phenotype promotes the resilience and stability of cancer cells. The senescence escape of polyploid cancer cells is associated with depolyploidization (neosis), a reverse process that can reprogram these cells to return to the mitotic cell cycle via epigenetic silencing of the cell cycle inhibitor, p21^WAF1^. These mitotically propagating, paradiploid-descendent tumor cells also exhibit a transcriptomic profile correlated with cytokine reprogramming. More specifically, the increased expression of pro-inflammatory cytokines, comprising IL-1beta, IL-6 and IL-8, strongly activates the TGF-beta pathway, resulting in the acquisition of EMT features, including a mesenchymal phenotype, as well as the upregulation of EMT markers [89,153,156]. These events are also accompanied by the upregulation of stemness markers [89,151,153,157,158] and tumorigenic and metastatic capacities, further suggesting a certain phenotypic stability and resilience of these daughter cells over time [109,151,159,160].

Collectively, these data demonstrate that polyploidy associated with TIS escape may play a central role in the survival of some cancer cells through the acquisition of a mesenchymal phenotype. This will lead to eventual clonogenic re-growth of a tumor following genotoxic stress induced by radio- or chemotherapy [76,87,151,152,161,162,163,164,165].

## 8. Cellular Plasticity Can Result from an Interplay between Senescent Cells and the Immune Component

Senescent cells are the only type of cells capable of producing the so-called “secretory phenotype” (SASP). However, through the paracrine action of SASP factors like TGF-beta and the inflammatory cytokine IL-6, senescent cells can impact various regulatory circuits that are common between the immune system and the EMT program [54,166,167,168]. While an increase in this paracrine effect is observed following loss of the tumor suppressor p53, paracrine activity is further supported by patterns of epigenetic reprogramming (e.g., novel methylation landscapes) that are memorized by cells, even after overcoming senescence, as well as by the sustenance of active transcriptomic profiles for senescence and SASP-related genes [77,99,107,109,169].

The paracrine action of SASP factors can indeed be pro-tumoral via its effect on EMT, given the widely demonstrated role of EMT in invasion and metastatic dissemination [99,122,170,171] (Figure 3). In this context, genotoxic stress from OIS or TIS leads to the generation of micronuclei and genomic instability, preceding the activation of an inflammatory secretory phenotype controlled by the cGAS/STING/NFKB pathway [105,107,108,109,172]. Activation of the inflammatory SASP enhances migration and invasion properties that promote metastasis [173,174,175]. In this way, a few senescent cancer cells can trigger an increase in the number of cancer cells despite chemo- or radiotherapeutic treatments [45,76,99,100,173,176,177,178,179,180,181]. Nevertheless, although reports remain merely correlative as to the causal action of EMT, stemness or intrinsic plasticity in chemoresistance [170], it has been shown that resistance to treatment is not so much related to stemness as quiescence, highlighting the contrasting properties of quiescent and therapy-resistant cancer stem cells [182,183,184,185].

Recent findings reveal an interplay between the activation of the SASP and the composition of the tumor microenvironment (TME). Specifically, the SASP can modulate the population of immunosuppressive cells in the TME. Immature monocytes (MDSCs) are typically recruited to a site of senescence. However, the maturation of these MDSCs into a tumor-inhibiting M1 state can be prevented by adjacent cancer cells through lactate production as a result of their glycolytic metabolism [142]. This results in an inability to recruit NK lymphocytes [143] or CD8+ T cells. NK cells play an important role in hindering the EMT of cancer cells during metastatic progression. This is due to an increased susceptibility of cancer cells to NK cytotoxicity associated with the EMT phenotype (e.g., lowered e-cadherin) [186]. Thus, through modifications to the TME, the absence of NK-mediated clearance ultimately creates an immune umbrella over senescent cancer cells and creates a permissive environment for cancer cell growth and metastatic progression via the enhancement of EMT [186,187,188] (Figure 2 and Figure 3).

Finally, beyond epithelial cancer cells, even adjacent normal fibroblasts can undergo senescence and fuel tumor progression [189]. Yet the data remains unclear as to the autocrine and/or paracrine action of the SASP as well as other SASP factors that could contribute to invasion and dissemination via increased angiogenesis [190].

## 9. Conclusions

Senescence is essential for the sustenance of physiological processes like embryonic development and wound healing. However, senescence can also have more prolonged, negative effects that contribute to deteriorative conditions like fibrosis and cancer. By definition, a senescent cell is a cell that has exited the cell cycle and therefore lacks proliferative capacity. Thus, following an oncogenic insult, the entry into senescence is considered an oncosuppressive process. Yet, through the accumulation of cell-autonomous features like genetic and epigenetic changes during the senescent period, an individual cell can escape from senescence while becoming plastic. Moreover, via cell non-autonomous features like the SASP, which has a paracrine mode of action, tumorigenic capacities can be imparted to entire cell populations. Hence, senescence escape as well as population-level reprogramming through the SASP are essential pathways for the progression of epithelial tumors. The acquisition of plasticity by EMT can also allow cells to completely avoid senescence. The propensity of a cell to bypass OIS is determined by the state of cellular differentiation and has a major impact on the genetic history of tumor development. Furthermore, the immune system is intricately involved in senescence-mediated pro-tumor effects. First, an increasing flow of senescent cells during tumor progression can embolize the immune system and alter the composition of the SASP according to the genetic background of cells, thus modulating immunoediting of these cells. Second, the presence of cancer cells can modify the tumor microenvironment, notably through the release of lactic acid, preventing the elimination by the immune system of senescent cancer cells that are already present or are emerging. Finally, it is important to distinguish the precancerous stage at which the NASP and the PASP regulate the elimination of potentially cancerous senescent cells. Similarly, the cancer stage at which the NASP allows for an increased plasticity and aggressiveness of tumor cells must also be determined. A potential and novel therapeutic avenue would be to target the NASP-mediated plasticity of tumor cells, with the aim of hindering the underlying migratory and metastatic properties.

## Figures and Tables

**Figure 1 cancers-13-04561-f001:**
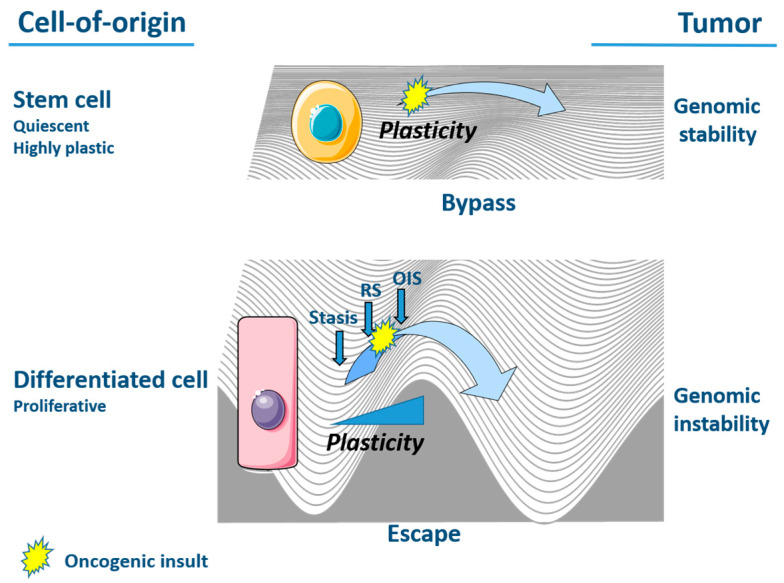
Dualistic model for tumor initiation. Stem cells are quiescent and present intrinsic EMT-dependent plasticity features. Stem cells are able to bypass senescence and give rise to genomically stable tumors [95]. Inversely, the escape from senescence of differentiated cells requires the acquisition of polyploidy and extensive cellular reprogramming, resulting in genomically rearranged tumors. Stasis: stress-associated senescence barrier. RS: replicative senescence. OIS: oncogene-induced senescence. This figure was made using elements from Servier Medical Art, which are licensed under a Creative Commons Attribution 3.0 Unported License: https://smart.servier.com (accessed on 17 August 2021).

**Figure 2 cancers-13-04561-f002:**
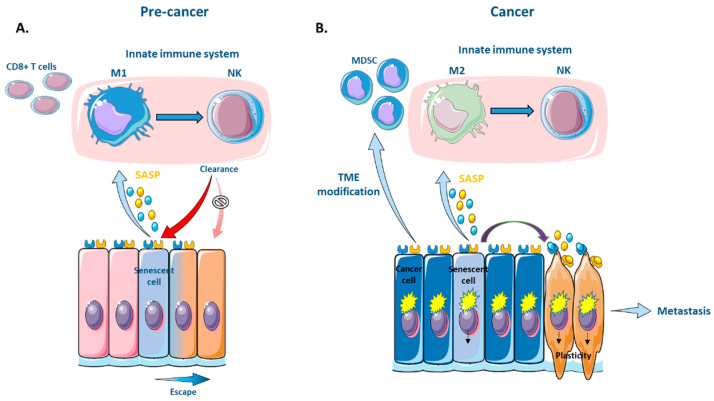
Immune evasion of senescent cells through acquisition of plasticity. (**A**) The acquisition of plasticity during senescence escape alters the secretory phenotype, leading to a redirection of macrophage polarization, preventing the clearance of senescent cancer cells [124]. EMT/plasticity modulates the expression of immune receptors (downregulated MHC-I) and the recognition of senescent cells by the immune system. (**B**) The SASP produced by senescent cancer cells leads to the recruitment of immature monocytes (MDSCs). Their maturation is prevented by adjacent cancer cells through metabolic modification of the tumor microenvironment (TME) [142]. The accumulation of MDSCs unable to polarize into M1, nor recruit NKs [143] or CD8+ T cells, in turn leads to the creation of an immune umbrella for cancer cell growth and will promote metastatic progression via the enhancement of EMT. This figure was made using elements from Servier Medical Art, which are licensed under a Creative Commons Attribution 3.0 Unported License: https://smart.servier.com (accessed on 17 August 2021).

**Figure 3 cancers-13-04561-f003:**
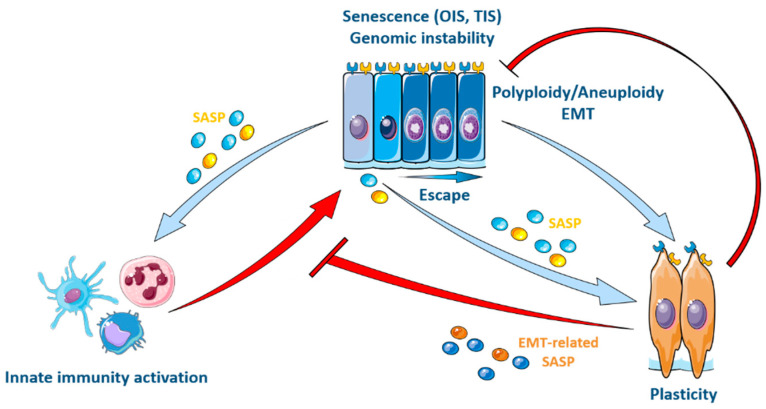
Plasticity acquired following senescence escape leads to the modulation of innate immunity. Senescence induced by different stimuli activates the production of an inflammatory secretory phenotype (SASP) under the control of the cGAS/STING/NFKB pathway. Its activation is required for the recruitment of immune cells and the clearance of senescent cancer cells by the innate immune system. On the other hand, SASP, together with polyploidy and EMT features acquired after escape from senescence, leads to the enhanced cancer cell plasticity associated with a deleterious secretory phenotype, allowing immune evasion of cancer cells. This figure was made using elements from Servier Medical Art, which are licensed under a Creative Commons Attribution 3.0 Unported License: https://smart.servier.com (accessed on 17 August 2021).

**Table 1 cancers-13-04561-t001:** Secreted factors in the NF-KB-dependent inflammatory-type SASP and p53-dependent SASP [103,110].

Type of SASP	Secreted Factors
**NF-KB-dependent inflammatory-type SASP**	IL1α, Ilβ, IL6, IL8, IL10, CXCL1, CXCL2, VEGF, MMP3, TNFα, FGF
**p53-dependent SASP**	GRO, GROα, IGFBP3, LIF, IL6, IL8, CCL2, CCL17, Leptin, ISG15, GDF15, TGFα

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
