# Peer review of "Cellular Plasticity: A Route to Senescence Exit and Tumorigenesis"

_cancers, 2021, doi:10.3390/cancers13184561_

Round 1
Reviewer 1 Report
Authors present a review on the features linking together senescence associated factors and their role in acquiring cellular plasticity in the context of cancer initiation and progression. The review is clear and concise highlighting key points related to senescence associated cellular plasticity mechanisms, which are backed up by relevant research literature. The proposed models for acquiring plasticity in tumor initiation and progression by overcoming senescence are also illustrated in the figures included in this review, providing additional clarity to the main text. The review could benefit from adding a few sentences about the role of TWISTs in overriding oncogene-induced senescence in the section 6 (reference 34 in the review). Additionally, a few minor corrections should be made.
- In line 110, there should be 10 to the power of 4 (positive number) but instead there is -4. This should be corrected.
- In figure 1 authors include reference 95, which is the same as reference 94 in the text, this needs to be corrected.
- Both in figure 2 and in the main text (section 8) authors use the term CD8+ LT but it is not mentioned what does the LT stand for? If authors refer to CD8-positive T lymphocytes, the use of more common abbreviation, CD8+ T cells, would be recommended to avoid confusion.
- In line 336 the word “thusly” should be replaced with a more appropriate expression “thus”.
Author Response
We greatly appreciate your review of our manuscript Cancers-1364399 entitled: " Cellular plasticity: a route to senescence exit and tumorigenesis".
Please find below our point-by-point replies to the reviewers’ comments.
Reviewers' comments:
Reviewer #1:
1.1. “Authors present a review on the features linking together senescence associated factors and their role in acquiring cellular plasticity in the context of cancer initiation and progression. The review is clear and concise highlighting key points related to senescence associated cellular plasticity mechanisms, which are backed up by relevant research literature. The proposed models for acquiring plasticity in tumor initiation and progression by overcoming senescence are also illustrated in the figures included in this review, providing additional clarity to the main text. The review could benefit from adding a few sentences about the role of TWISTs in overriding oncogene-induced senescence in the section 6 (reference 34 in the review).
We thank the reviewer for this comment. We have added additional remarks on the role of TWIST in overriding oncogene-induced senescence (line 189-195).
Additionally, a few minor corrections should be made.
1.2. In line 110, there should be 10 to the power of 4 (positive number) but instead there is -4. This should be corrected.”
We have corrected this typing mistake in line 113 in the revised manuscript
1.3. “In figure 1 authors include reference 95, which is the same as reference 94 in the text, this needs to be corrected.”
We have corrected the duplication of references 94-95. All references have been updated.
1.4. “Both in figure 2 and in the main text (section 8) authors use the term CD8+ LT but it is not mentioned what does the LT stand for? If authors refer to CD8-positive T lymphocytes, the use of more common abbreviation, CD8+ T cells, would be recommended to avoid confusion.”
We thank the reviewer for this recommendation. We have replaced the term CD8+LT cells with the more common abbreviation CD8+ T cells.
1.5. “In line 336 the word “thusly” should be replaced with a more appropriate expression “thus”.”
We have replaced “thusly” with “In this context” in line 367 in the revised manuscript.
Reviewer 2 Report
De Blander and colleagues revise extensively the concept of cell plasticity correlated to the senescence contest. I found this review interesting and well written. Concepts are supported by literature and well resumed. In my opinion, only minor revisions should be required.
Listed below are my observations:
Pag. 2 lines 77-79: should be revised and written in a more simple version
Pag. 3 lines 99-101: Simplify the concept
Figure 1 appears not immediate, it should be improved
It should be useful to the reader (and it should enrich the review) to add a Table in which the NASP and PASP factors are listed
Author Response
We greatly appreciate your review of our manuscript Cancers-1364399 entitled: " Cellular plasticity: a route to senescence exit and tumorigenesis".
Please find below our point-by-point replies to the reviewers’ comments.
Reviewers' comments:
Reviewer #2:
2.1. “De Blander and colleagues revise extensively the concept of cell plasticity correlated to the senescence contest. I found this review interesting and well written. Concepts are supported by literature and well resumed. In my opinion, only minor revisions should be required.
Listed below are my observations:
Pag. 2 lines 77-79: should be revised and written in a more simple version”
We thank the reviewer for this remark. We propose the revised sentence: “These cells, reprogrammed during chronic inflammation, are a key component of the senescence response and their aberrant accumulation further catalyzes the generation of damaged fibrotic tissue.”
2.2. Pag. 3 lines 99-101: Simplify the concept
We agree with the reviewer’s suggestion. We have revised the concept as follows: “However, some cells like epithelial cells are capable of spontaneously reverting to a proliferative state following cell cycle arrest after encountering a stress signal, process known as “senescence escape”. Senescence escape is facilitated by altering the activity of chromatin regulators, metabolic pathways or extracellular pH.” (lines 96-105 in the revised manuscript).
2.3. Figure 1 appears not immediate, it should be improved
We thank the reviewer for his valuable comment. We aimed to illustrate the dualistic aspect of tumor initiation. As suggested by the reviewer, we have introduced Figure 1 earlier in the section (line 203 in the revised manuscript).
2.4. It should be useful to the reader (and it should enrich the review) to add a Table in which the NASP and PASP factors are listed”
As suggested by the reviewer, we have included a table in which the NASP and PASP factors are listed (reference in line 235 and Table 1 in line 265 in the revised manuscript).